# Patisiran Enhances Muscle Mass after Nine Months of Treatment in ATTRv Amyloidosis: A Study with Bioelectrical Impedance Analysis and Handgrip Strength

**DOI:** 10.3390/biomedicines11010062

**Published:** 2022-12-27

**Authors:** Vincenzo Di Stefano, Ewan Thomas, Paolo Alonge, Valerio Giustino, Guglielmo Pillitteri, Ignazio Leale, Angelo Torrente, Antonia Pignolo, Davide Norata, Salvatore Iacono, Antonino Lupica, Antonio Palma, Giuseppe Battaglia, Filippo Brighina

**Affiliations:** 1Neurology Unit, Department of Biomedicine, Neuroscience, and Advanced Diagnostics (BiND), University of Palermo, 90129 Palermo, Italy; 2Department of Psychology, Educational Science and Human Movement, University of Palermo, 90127 Palermo, Italy; 3Neurological Clinic, Department of Experimental and Clinical Medicine, Marche Polytechnic University, 60121 Ancona, Italy

**Keywords:** hereditary amyloid neuropathy, ATTRv, TTR, handgrip strength, bioelectrical impedance analysis, patisiran

## Abstract

Background and aims. Hereditary transthyretin amyloidosis with polyneuropathy (ATTRv) is caused by mutations in the TTR gene, leading to misfolded monomers that aggregate generating amyloid fibrils. The clinical phenotype is heterogeneous, characterized by a multisystemic disease affecting the sensorimotor, autonomic functions along with other organs. Patisiran is a small interfering RNA acting as a TTR silencer approved for the treatment of ATTRv. Punctual and detailed instrumental biomarkers are on demand for ATTRv to measure the severity of the disease and monitor progression and response to treatment. Methods. Fifteen patients affected by ATTRv amyloidosis (66.4 ± 7.8 years, six males) were evaluated before the start of therapy with patisiran and after 9-months of follow-up. The clinical and instrumental evaluation included body weight and height; Coutinho stage; Neuropathy Impairment Score (NIS); Karnofsky performance status (KPS); Norfolk QOL Questionnaire; Six-minute walking test (6 MWT); nerve conduction studies; handgrip strength (HGS); and bioimpedance analysis (BIA). Results. Body composition significantly changed following the 9-months pharmacological treatment. In particular, the patients exhibited an increase in fat free mass, body cell mass, and body weight with a decrease in fat mass. A significant increase after 9 months of treatment was observed for the 6 MWT. Coutinho stage, KPS, NIS, NIS-W, nerve conduction studies, Norfolk, COMPASS-31 scale, and HGS remained unchanged. Conclusions. BIA might represent a useful tool to assess the effects of multiorgan damage in ATTRv and to monitor disease progression and response to treatments. More evidence is still needed for HGS. Patisiran stabilizes polyneuropathy and preserves motor strength by increasing muscle mass after 9 months of treatment.

## 1. Introduction

Hereditary transthyretin amyloidosis with polyneuropathy (ATTRv) is an adult-onset multisystemic disease, affecting the sensory-motor and autonomic functions along with other organs, especially the heart, gastrointestinal tract, eyes, and kidney [1]. ATTRv has an autosomal dominant pattern of inheritance, and worldwide, its global prevalence is estimated at up to 38,000 people [2,3]. TTR is a transport protein in the serum and cerebrospinal fluid that carries the thyroid hormone thyroxine and retinol-binding protein bound to retinol. It is encoded by the *TTR* gene located in the 18th chromosome. The TTR protein forms tetramers constituted by monomers rich in a beta sheet structure [4]. The presence of missense mutations conducts to a less stable tetramer by altering the amino acid sequence, thus favoring its dissociation. The misfolded monomers aggregate, generating amyloid fibrils, which precipitate into tissues [4,5]. The most frequent TTR mutation is V30M with an early onset (<50 years) in endemic areas, whereas a late onset (>50 years) phenotype is prevalent in non-endemic areas [6,7,8]. In peripheral nerves, amyloid fibrils cause a rapidly progressive peripheral sensory-motor polyneuropathy with significant disability, while in the heart, they generate a cardiomyopathy that may influence the progression of the disease and survival of ATTRv patients [9]. Nowadays, several available treatment options are effective in early disease stages of ATTRv [10]. In particular, patisiran, a small interfering RNA acting as a TTR silencer, approved in Italy in 2020, has been shown to stabilize the course of polyneuropathy in ATTRv, but its administration is bound to the presence of neuropathy as a manifestation [11,12]. Consequently, a meticulous assessment of neuropathy is essential. According to the most common scale to assess the overall burden of polyneuropathy [13], ATTRv evolves into three successive stages: in the first, patients have a sensory polyneuropathy leading to difficulty in walking without assistance; in the second phase, there is a significant limitation in ambulation; finally, patients become wheelchair-bound or bedridden. Additionally, in most studies, neuropathy is assessed through nerve conduction studies, Norfolk QOL-DN questionnaires, NIS + 7 scale, and six-minute walking test (6 MWT) [13,14]. Some researchers are involved in the evaluation of serum biomarkers of neuropathy such as light chain neurofilaments [15]. Of interest, muscle strength can be measured through handgrip tools [16]. The handgrip strength (HGS) test evaluates the force that a person is able to produce when grasping an object. Since HGS is performed with just a dynamometer, it is an easy and cost-effective tool that allows one to measure the maximum isometric handgrip strength of the hand and forearm muscles [17]. HGS provides a quantitative measure of distal strength, and several studies have confirmed that HGS can be used as a diagnostic and prognostic tool in several chronic diseases such as sarcopenia [18] we well as acquired neuropathy [19] such as carpal tunnel syndrome (CTS) [20] and hereditary neuropathy such as Charcot–Marie–Tooth [21]. Apart from a progressive sensory-motor neuropathy and cardiomyopathy, accompanying symptoms are reported in ATTRv such as autonomic dysfunction as well as gastrointestinal problems and unexplained weight loss [22]. These self-reported symptoms are difficult to demonstrate and consequently often underestimated [23]. Some questionnaires such as the COMPASS-31 scale and CADT are commonly used in clinical practice, but an instrumental evaluation of dysautonomia through neurophysiological instruments is not systematically performed at most centers [13]. Bioelectrical impedance analysis (BIA) is a very sensitive tool to examine the composition of the body tissues in polyneuropathies and conditions with dysautonomia. Hence, BIA might accurately estimate the body composition in terms of muscle and fat masses as well as water content in ATTRv patients; this might be an indirect measure of dysautonomia and gastrointestinal function. Although BIA has been used to evaluate sarcopenia in diabetic neuropathy and AL amyloidosis [24,25], to our knowledge, there have been no studies to have assessed body composition through bioimpedance analysis in ATTRv patients. Taken together, HGS and BIA might have a potential in the assessment of the severity of the disease and the beneficial effects of treatments. However, clinical trials and real-world studies have demonstrated the effect of patisiran on PND score, NIS scale, and questionnaires [26], but there are no instrumental data on muscle strength, body composition, and gastrointestinal symptoms. Of note, the use of instrumental biomarkers for the evaluation of tissue damage in ATTRv might lead to a higher sensitive and specific approach; moreover, these biomarkers might contribute to detecting the exact clinical onset of the disease in carriers of TTR mutation, giving them the opportunity to be treated early when the polyneuropathy starts. In this study, instrumental data by means of handgrip strength and BIA were systematically collected in ATTRv patients at the baseline and after 9 months of follow-up to evaluate their potential to measure the pathophysiological alterations of ATTRv amyloidosis and the effects of therapy with patisiran.

## 2. Materials and Methods

### 2.1. Study Procedures

In this prospective study, we described the hand strength assessed by handgrip evaluation and body composition assessed through bioelectrical impedance analysis in a cohort of ATTRv patients treated with patisiran to detect early modification and response to therapy at 9 months of follow-up. The study was conducted in accordance with the Declaration of Helsinki and approved by the Ethics Committee Palermo I, University of Palermo (Protocol code 07/2020; 13 July 2020). Informed consent was obtained from all subjects involved in the study.

### 2.2. Patient Demographics and Clinical Features

Fifteen patients affected by ATTRv amyloidosis (66.4 ± 7.8 years, six males) participated in the study. Genetic testing confirmed a mutation in heterozygosis in the *TTR* gene in all patients enrolled. F64L (p.F84L) mutation was encountered in 11 patients, followed by E89Q (p.E109Q), V122I (p.V142I), H90A (p.H110A), and S77F (p.S97F) in one patient, respectively. The most frequent symptoms were carpal tunnel syndrome (80%), gastrointestinal disturbances (60%), ataxia (50%), and weight loss and autonomic dysfunction (45%). The anthropometric characteristics of the patients are shown in Table 1.

All patients were evaluated at the time of starting therapy with patisiran and after 9-months of follow-up. The treatment with patisiran was scheduled as per the therapeutic protocol (0.3 mg per kilogram of body weight once every 3 weeks). The clinical and instrumental evaluation included body weight and height; Coutinho stage; Neuropathy Impairment Score (NIS); Karnofsky performance status (KPS); Norfolk QOL Questionnaire; six-minute walking test (6 MWT); nerve conduction studies (NCS); handgrip measures; and bioimpedance analysis.

### 2.3. Coutinho Stage

Familial amyloidotic polyneuropathy (FAP) staging is based on the walking ability of the patients: stage 0 is for asymptomatic patients; stage I identifies patients with symptoms but unimpaired ambulation; stage II patients require assistance for walking; stage III patients are wheelchair-bound or bedridden [13].

### 2.4. Neuropathy Impairment Score (NIS)

NIS evaluates the strength, reflexes, and sensation [13]. It is one of the most well-known clinical instruments to evaluate and quantify the burden of neuropathy. It ranges from 0 to 192 points, with higher scores indicating a worse impairment. We calculated the total score (NIS, muscle weakness, muscle stretch reflexes, and sensation) and the motor component alone for the whole body (NIS-W, muscle weakness).

### 2.5. Karnofsky Performance Status (KPS)

The KPS score was used to quantify the subjects’ ability to perform normal daily life activities and their need for assistance (ranging from 0 [dead] to 100 [normal; no complaints]) [27].

### 2.6. Norfolk QOL Questionnaire

The Norfolk QOL [28] is a 35-item questionnaire designed to evaluate the impact on the patient’s life of symptoms related to neuropathy, ranging from −2 (best QOL) to 138 (worst QOL). It has been used to assess the quality of life in patients affected by ATTRv amyloidosis treated with tafamidis [29], inotersen [30], patisiran [31], and vutrisiran [32].

### 2.7. Six-Minute Walking Test (6MWT)

The 6MWT is an easy clinical test that estimates the patient’s performance in daily activities; it is a measurement used to assess the functional capacity (i.e., the aerobic endurance) [33]. Although it has been extensively used for cardiopulmonary diseases, it has only been used anecdotally on familial amyloidotic neuropathy [34]; however, Vita et al. proved its reliability in the evaluation of ATTRv patients with neuropathy but without cardiovascular involvement [35]. The 6MWT was conducted by measuring the distance covered in a period of 6 min by the patient walking quickly on a flat, hard surface [36]. Each participant was asked to walk as much as possible for 6 min in a flat corridor where two cones, placed 30 m from each other, marked the turning points. At the verbal command of the researcher, each participant, placed upright next to one of the two cones, began to walk to the other cone, and then back, and so on. The distance walked in 6 min measured in meters was recorded (i.e., the 6-min walk distance).

### 2.8. Nerve Conduction Studies (NCSs)

NCSs were performed in both median nerves for all subjects enrolled according to standard procedures (i.e., bipolar surface stimulating electrodes delivering rectangular pulses 0.1–0.5 ms in duration and recording electrodes placed over the recording site with a ground electrode placed between recording and stimulation electrodes) [37].

### 2.9. Handgrip Test

Each participant carried out the handgrip test as recommended by the American Society of Hand Therapists [38], that is, in a sitting position with their back leaning against the backrest of the chair and elbow joint positioned at a 90° angle. At the verbal command of the researcher, each participant had to tighten the handle of a mechanical dynamometer (KernMap model 80 K1—Kern^®^, Kern&Sohn GmbH, Balingen, Germany), exerting their maximum isometric handgrip strength for 3 s. Each participant performed three trials both with the dominant and the non-dominant hand with 3 min rest between trials. The best trial was considered for statistical analyses [39].

### 2.10. Bioelectrical Impedance Analysis (BIA)

BIA is a non-invasive, radiation-free examination that evaluates body composition using a device that allows one to estimate the quality of different tissues (e.g., fat and muscle) since it is able to measure the different electrical conductivity of the tissues. In detail, the body is crossed by a low voltage alternating current generated by the device which, in turn, measures the impedance (resistance and reactance of the tissues) [40]. BIA allows one to estimate body cell mass (BCM), extracellular water (ECW), fat mass (FM) and fat-free mass (FFM). Prior to the BIA evaluation, the age and weight of each participant were entered into the software (Bodygram, Akern; Montacchiello, Pisa, Italy). Then, each participant was asked to lie down on a cot and the surface electrodes were applied on the right-hand and right-foot and the device (BIA 101, Akern; Montacchiello, Pisa, Italy) measured the body composition.

### 2.11. Statistical Analysis

Data were presented as means ± standard deviations. The Shapiro–Wilks test was used to detect normality among the distribution of variables. If the data presented a normal distribution, a paired t-test was used to detect the differences between the baseline and post-9-month treatment of the included patients. If the variables were not normally distributed, a Wilcoxon signed rank test was adopted instead. Subgroup analysis concerning gender was carried out on the significant parameters. Absolute differences between the post- and pre-values were calculated. Pearson’s correlation was finally carried out to identify associations among the performance and clinical variables. Significance was set at 0.05 for all analyzed variables. All analyses were carried out with Jamovi (The Jamovi project (2021), Jamovi (Version 1.8.0.1) [Computer Software]. Retrieved from https://www.jamovi.org, accessed on 1 December 2022).

## 3. Results

### 3.1. Clinical and Neurophysiological Evaluation

Table 2 shows the clinical data of 15 ATTRv patients included in the study at the baseline and after 9-months of follow-up.

Twelve patients presented mild symptoms (FAP stage 1) and three patients had moderate-severe symptoms (FAP stage 2). Among the clinical variable and scales included, the FAP stage, KPS, NIS, NIS-W, Norfolk, and COMPASS-31 scale resulted in being unchanged following 9 months of treatment with patisiran. Nerve conduction studies demonstrated axonal neuropathy in all subjects in the absence of significant differences between the baseline and follow-up evaluations. However, a significant increase after 9 months of treatment was observed for the 6MWT (229.6 ± 72.6 vs. 260.9 ± 69.8 m, *p* = 0.033).

### 3.2. Bioelectrical Impedance Analysis (BIA)

Body composition significantly changed following the 9-months pharmacological treatment. Figure 1 shows the data FM, FFM, and BCM at the baseline and after 9 months of treatment with patisiran. In particular, the patients exhibited an increase in FFM (pre 48.8 ± 11.9 kg vs. post 52.0 ± 12.1 kg, *p* = 0.005), which also translated to an increase in BCM (pre 23.1 ± 7.5 vs. post 26.4 ± 7.8 kg, *p* = 0.014). Conversely, a significant decrease in FM (pre 21.5 ± 9.8 vs. post 20.7 ± 8.5 kg, *p* = 0.012) was observed. Overall, this led to an increase in body weight from 70.3 ± 19.8 to 73.1 ± 21.1 kg (*p* = 0.044). No significant differences in TBW and ECW were observed.

### 3.3. Handgrip Strength

Regarding the performance measures, despite increased muscle mass being observed, no significant differences were evaluated for measures of the upper limbs, neither for the right nor the left limb. Details of the performance measures are shown in Table 2.

Differences between the male and female participants were also calculated for all of the above-mentioned variables. No significant gender difference was observed for any of the analyzed variables. Correlational analysis detected that significant and meaningful associations were present among the 6 MWT and BCM (r = 0.63, *p* = 0.012), the 6 MWT and the Norfolk scale (r = −0.76, *p* = 0.001), and the 6 MWT and both HG (r = 0.61, *p* = 0.016, right and r = 0.60, *p* = 0.016 left). In addition, the Norfolk scale was also negatively correlated with BCM (r = −0.61, *p* = 0.016), and both HG (r = −0.69, *p* = 0.005, right and r = −0.68, *p* = 0.004 left), respectively, while BCM was also positively associated with the HG measures (r = 0.79, *p* = 0.001, right and r = 0.85, *p* = 0.001 left).

## 4. Discussion

The principal finding of the present study was a significant increment in body weight accompanied by relevant changes in body composition in ATTRv patients 9 months after the start of treatment with the RNA silencer patisiran. In more detail, there was a more pronounced increment on fat-free mass (FFM) and body cell mass (BCM), which are both the expression of muscle mass; also, the increase in muscle mass was accompanied by a reduction in the fat mass (FM) without any modification in the water content. Moreover, as most patients presented unexplained weight loss before treatment, the weight gain was characterized by an increment in muscle mass instead of fat. This finding might support a role for patisiran in the reorganization of motor units in hypotrophic muscles, since a reduction in amyloid deposition might interrupt axonal damage, favoring reinnervation and neurotrophic processes. Moreover, the benefits on the patient health cannot be explained by the stabilization of neuropathy alone: an improvement in gastrointestinal manifestations might also have guaranteed a better absorption of micronutrients, thus solving a deficiency status. These findings are in line with the preservation of residual motor strength demonstrated by clinical scales (stable FAP stage and NIS) and HGS and the beneficial effect on 6 MWT. Of interest, the 6MWT showed a significant improvement in patient autonomy: the mean walked distance increased by 31.3 m after treatment, even if there were no significant changes in the upper limb strength (however, it should be noted that the 6MWT showed a correlation with both HGS). On this regard, it should be considered that HGS assesses the strength in upper limbs, while the 6MWT evaluates the global ambulation process; hence, an increase in walking distance might not be related to an improvement in strength alone, as balance and coordination also play a fundamental role. Moreover, the correlation between the 6MWT and BCM suggests that the increase in functional capacity confirmed by the 6MWT could depend not only on the stabilization of the neuropathy, but also on the improvement in the absorption of nutrients and the subsequent increase in the metabolic capacity of muscles.

HGS is a simple and cheap tool that is able to evaluate the distal strength in several conditions including hereditary polyneuropathies [21]. Moreover, Anbarasan et al. demonstrated how it represents a functional outcome measure that significantly improved after mini carpal tunnel release [20]. A previous study from our group proved that HGS is reduced in ATTRv patients, probably because of bilateral carpal tunnel syndrome and polyneuropathy [16]. Furthermore, HGS showed a negative correlation with NIS scores, while it was positively correlated with the neurophysiological evaluation of the median compound motor action potential amplitude [16]. In the present study, despite an increase in the muscle mass and a positive correlation with BCM (r = 0.79, *p* = 0.001, right and r = 0.85, *p* = 0.001 left), HGS did not show any significant variation after 9 months of therapy. However, this finding is encouraging and may account for a preservation of the patient’s distal strength. Furthermore, HGS was positively correlated with the 6MWT (r = 0.61, *p* = 0.016, right and r = 0.60, *p* = 0.016 left), suggesting that HGS might be related to exercise capacity, as already shown in different conditions [41]. Finally, the Norfolk QOL scale and HGS presented a negative correlation (r = −0.69, *p* = 0.005, right and r = −0.68, *p* = 0.004 left), supporting a relationship between HGS and the patient’s quality of life.

A not-significant reduction in the Norfolk QOL mean scores (−3.5) was reported in our population, with differed to the significant reduction after 18 months of treatment found on the APOLLO trial population (−6.7 ± 1.8). However, real-life data support a worsening of Norfolk QOL scores in the first 6 months of treatment with an improvement after 12 months [26]. Our observations show that a trend of improvement could be seen after 9 months of treatment, confirming that there might be a latency period before observing the benefits with Norfolk QOL. Of note, these findings suggest that BIA might estimate the overall functional capacity of the patient detecting improvements better than the clinical scales.

This study had some limitations. The selection bias and the low number of patients might have led to interpreting errors. Additionally, the evaluation of upper limbs on HGS might have carried an underestimation of the benefits on lower limbs. We also did not use a modified BMI (considering the serum albumin levels), which is a useful biomarker in malnutrition status. Future studies are needed to validate the use of these promising tools in clinical practice.

## 5. Conclusions

Punctual and detailed instrumental biomarkers are in demand for ATTRv to measure the severity of the disease burden and to monitor progression and/or response to treatment. Indeed, the instrumental evaluation of ATTRv patients might detect mild changes that may often be underrecognized by clinical evaluation alone. The handgrip test and bioelectrical impedance analysis are rapid, simple, and non-invasive tools that can be easily performed and do not require expensive equipment. Moreover, BIA might represent useful tools to assess the effects of multiorgan damage in ATTRv and to monitor disease progression and response to treatments. More data are still needed for HGS. Our data denote that patisiran stabilizes polyneuropathy and preserves motor strength by increasing the muscle mass after 9 months of treatment. We believe that HGS and BIA might find an application in clinical practice due to their low cost and high reliability. Further studies are needed to confirm our findings and clarify a possible role of such tools in the diagnostic process.

## Figures and Tables

**Figure 1 biomedicines-11-00062-f001:**
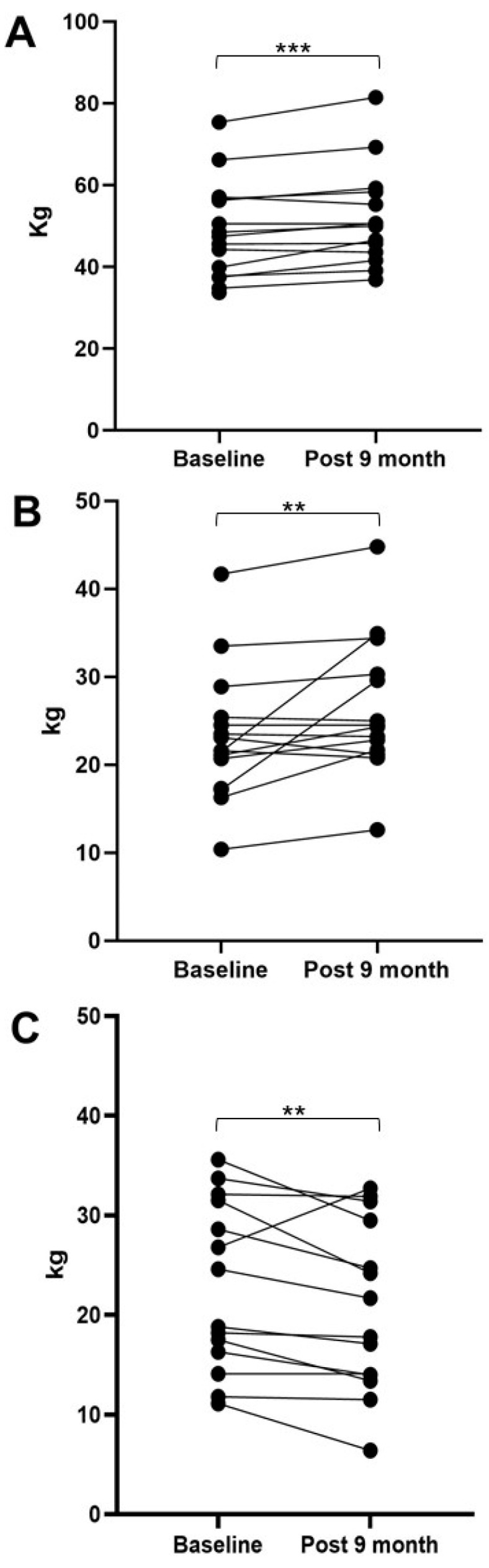
Body composition at the baseline and after 9 months of treatment with patisiran. Panel (**A**) represents individual patient data for FFM: free fat mass; Panel (**B**) represents the individual patient data for BCM (body cell mass); Panel (**C**) represents the individual patient data for FM (fat mass). ** *p* < 0.05; *** *p* < 0.01.

**Table 1 biomedicines-11-00062-t001:** Descriptive characteristics of the included patients.

	Baseline	Post 9 Month	*p*
Age (years)	66.4 ± 7.8	66.9 ± 7.7	ns
Height (cm)	163.0 ± 10.9	163.0 ± 10.9	ns
Weight (kg)	70.3 ± 19.8	73.1 ± 21.1	0.044

Data are presented as means ± st.dv; ns not significant.

**Table 2 biomedicines-11-00062-t002:** Body composition, strength, and 6-min walking test outcomes.

	Baseline	Post 9 Month	*p*
Clinical evaluation			
FAP stage	1.13 ± 0.5	1.13 ± 0.5	ns
Karnofski performance status	72.7 ± 13.8	75.3 ± 14.1	ns
NIS	30.9 ± 29.2	31.4 ± 25.6	ns
NIS-W	14.6 ± 17.8	14.7 ± 14.5	ns
Norfolk	51.0 ± 31.6	47.8 ± 29.5	ns
COMPASS-31	18.9 ± 9.1	19.8 ± 9.2	ns
6MWT (m)	229.6 ± 72.6	260.9 ± 69.8	0.033
Body Composition			
FFM (kg)	48.8 ± 11.9	52.0 ± 12.1	0.005
BCM (kg)	23.1 ± 7.5	26.4 ± 7.8	0.014
FM (kg)	21.5 ± 9.8	20.7 ± 8.5	0.012
TBW (l)	36.5 ± 9.2	38.4 ± 9.1	ns
ECW (l)	19.0 ± 4.9	18.7 ± 4.5	ns
Strength			
HG R	20.6 ± 13.8	21.4 ± 13.9	ns
HG L	22.5 ± 11.8	22.1 ± 10.9	ns

Data are presented as the means ± st.dv; ns not significant. Ns = not significant. FAP: familial amyloidotic polyneuropathy; NIS: neuropathy impairment score; NIS-W: neuropathy impairment score muscle weakness; 6MWT: 6-min walking test; BCM: body cell mass; ECW: extra cellular water; FFM: free fat mass; FM: fat mass; HG: handgrip; L: left; R: right; TBW: total body water.

## Data Availability

Data are available from the corresponding author upon reasonable request.

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
