# Peer review of "Patisiran Enhances Muscle Mass after Nine Months of Treatment in ATTRv Amyloidosis: A Study with Bioelectrical Impedance Analysis and Handgrip Strength"

_biomedicines, 2022, doi:10.3390/biomedicines11010062_

Round 1
Reviewer 1 Report
This is a very small study without controls for drug therapy. Hence its relevance can best be analyzed as a collection of "n" of 1 trials, with the data presented as the change in each patient rather than as the mean or median of the values of the entire group, particularly the BIA values. You are arguing that BIA may be a more sensitive marker of amyloid polyneuropathy than what are presently the conventional markers. Nonetheless you show statistically significant findings in the conventionally used 6MWT and body weight. To argue that BIA was more sensitive you would need significant changes in values in BIA earlier than those seen in weight or 6MWT. If you have those your case would be much stronger. Given the changes in weight and the relevance of mBMI in measuring responses to therapy I am surprised that you did not include such measurements in your patients. Again, it would have tested your hypothesis that BIA is a more sensitive measure. It is a bit disconcerting that the mechanism underlying the observed change in BIA does not quite fit with the prior data from AL, in which some of the changes were attributed to fluid overload related to cardiac or renal compromise. That study also noted differences in BIA related to the sex of the patients, males being greater than females. The sex distribution of your subjects is not noted. I do not think the data as presented support the conclusion that patisiran stabilizes polyneuropathy and preserves motor strength by increasing muscle mass. Nor have you shown that BIA is more sensitive than current parameters, i.e. 6MWT, nor that HGS adds to the current battery of tests utilized in assessing patients with FAP.
Author Response
Dear Editors and Reviewers,
thanks for your comments. We would like you will evaluate our revised version of the manuscript for possible publication in Biomedicines.
Reviewer #1:
1) This is a very small study without controls for drug therapy. Hence its relevance can best be analyzed as a collection of "n" of 1 trials, with the data presented as the change in each patient rather than as the mean or median of the values of the entire group, particularly the BIA values.
Thank you for this precious comment. The sample size has been underlined in the limitations. We have modified figure 1 presenting differences for each patient replacing histograms with graphs reported in the new figure 1. We hope that this presentation might be more informative and useful for the reviewer.
“Figure 1. Body composition at baseline and after 9 months of treatment with patisiran. Panel A represents individual patiens data for FFM: Free Fat Mass; Panel B represents individual patients’ data for BCM: Body Cell Mass; Panel C represents individual patients’ data for FM: Fat Mass. **p<0.05; ***p<0.01.”
2) You are arguing that BIA may be a more sensitive marker of amyloid polyneuropathy than what are presently the conventional markers. Nonetheless you show statistically significant findings in the conventionally used 6MWT and body weight. To argue that BIA was more sensitive you would need significant changes in values in BIA earlier than those seen in weight or 6MWT. If you have those your case would be much stronger.
Thank you for the valuable comments and the possibility to improve our paper. We partially agree with the reviewer, because ATTRv is a very rare disease, and 15 patients are not so low number in consideration of the rarity of the disease. Hence these data may be precious because BIA and HG measures have never been reported till now. Also, we reported statistically significant differences that strengthen these results even if the number of patients is low. Moreover, the aim of the study is not to assess the efficacy of patisiran in ATTRv patients, as this has already been demonstrated in randomized-controlled trials (APOLLO and Global-OLE), but we aimed to detect instrumental changes which might be more easily performed than clinical scales. Our findings have not the object to validate these methods, but they are preliminary results which may open the road to wider multicentre studies.
3) Given the changes in weight and the relevance of mBMI in measuring responses to therapy I am surprised that you did not include such measurements in your patients.
We did not use mBMI (considering albumin in the sera), because these are preliminary results from follow-up data. We added this on the limitations:
“Also, we did not use mBMI (taking into account the serum albumin levels), which is a useful biomarker in malnutrition status.”
4) Again, it would have tested your hypothesis that BIA is a more sensitive measure. It is a bit disconcerting that the mechanism underlying the observed change in BIA does not quite fit with the prior data from AL, in which some of the changes were attributed to fluid overload related to cardiac or renal compromise.
We thank the reviewer for this consideration. However, AL and ATTRv are two different conditions from several points of view. First, AL presents higher rated of cardiomyopathy and renal impairment which usually led to fluid accumulation or loss. Conversely, total water and extracellular water do not show significant differences, therefore there was no fluid retention that caused an increase in lean body mass, but it was the increase in BCM without an increase in fluids that caused an increase in FFM. Indeed, ATTRv amyloidosis is less frequently accompanied by cardiac and renal impairment when mutations are different from Val30Met, which is our case as the most common mutation was Phe64Leu, a typical neuropathic phenotype. This point alone might explain differences reported in BIA. On this perspective, changes detected from BIA are imputed to dysautonomia and gastrointestinal impairment rather than cardiac impairment in our cohort.
5) That study also noted differences in BIA related to the sex of the patients, males being greater than females. The sex distribution of your subjects is not noted.
The sex distribution of our subjects is reported in the abstract, results (paragraph 2.2). However, there were not significant differences depending on gender. Subgroup analysis concerning gender was carried out on significant parameters. Absolute differences between post and pre-values have been calculated. For a better evaluation we added these in the results, and we provide the following table for review-only use:
|
Table. Variation between Pre-post measures for body composition, strength and 6-minute walking test outcomes stratified by gender |
|||
|
|
ΔMale |
ΔFemale |
p |
|
Body Composition |
|
|
|
|
FFM (kg) |
2 |
3.2 |
ns |
|
BCM (kg) |
3 |
3 |
ns |
|
FM (kg) |
-1.5 |
-0.5 |
ns |
|
TBW (l) |
0.8 |
1.9 |
ns |
|
ECW (l) |
-1.2 |
-0.1 |
ns |
|
Strength |
|
|
|
|
HG R |
1 |
0.8 |
ns |
|
HG L |
1.2 |
-1.4 |
ns |
|
6MWT (m) |
27.3 |
36.3 |
ns |
|
Data are presented as absolute differences of mean pre and post measures; ns not significant. 6MWT: 6-minute walking test; BCM: Body Cell Mass; ECW: Extra Cellular Water; FFM: Free Fat Mass; FM: Fat Mass; HG: Hand Grip; L: Left; R: Right; TBW: Total Body Water. |
|||
We have also added in the text the following statements:
“Subgroup analysis concerning gender was carried out on significant parameters. Absolute differences between post and pre-values have been calculated.”
“Differences between male and female participants were also calculated for all above-mentioned variables. No significant gender difference was observed for any analysed variable.”
6) I do not think the data as presented support the conclusion that patisiran stabilizes polyneuropathy and preserves motor strength by increasing muscle mass. Nor have you shown that BIA is more sensitive than current parameters, i.e. 6MWT, nor that HGS adds to the current battery of tests utilized in assessing patients with FAP.
We have never stated that “BIA is more sensitive of 6MWT” (we doublechecked throughout the text and it was never affirmed nor in the abstract, conclusions and discussion): our data simply show that BIA might be able to detect relevant changes in body composition and we believe that HGS and BIA might find an application in clinical practice due to their low cost and high reliability. Also, BIA detected changes that went underrecognized by COMPASS-31 and other scales. Further studies are needed to confirm our findings and clarify a possible role of such tools in the diagnostic process.
Hoping in positive feedback we look forward to hearing from you soon.
Kind regards,
Vincenzo Di Stefano

Reviewer 2 Report
The authors conducted bioelectrical impedance analysis in 15 patients with hereditary transthyretin (ATTRv) amyloidosis before and 9 months after the administration of patisiran, one of the novel RNA interfering agents. The body composition significantly changed following 9 months of treatment.
This is an important study providing important insights into current knowledge on the treatment of ATTRv amyloidosis. Taking up the topic of ATTRv amyloidosis is timely because novel disease-modifying therapies, such as transthyretin stabilizers, RNA interfering agents, antisense oligonucleotides, and gene editing agents, now appear one after another. Therefore, this manuscript will attract broad range of readers. It is well written, and I do not have any critical comments.
Minor issues to strengthen this manuscript are raised as follows:
1. Please reconfirm the use of abbreviations. For example, “HGS” in the abstract should be spelled out.
2. Although the authors stressed hand grip strength for the monitoring the disease progression and response to treatment as well as bioelectrical impedance analysis, the values did not significantly change after the treatment. It seems to be better to tone down the issue of hand grip strength because the results of bioelectrical impedance analysis are sound.
3. Is this a retrospective study, or a prospective one?
4. I would suggest briefly stating the ethics issue in the methods section although it is incorporated in the last part of the main text.
5. The authors mentioned the issue of early-onset Val30Met cases from endemic foci and late-onset cases from non-endemic areas in the first paragraph of the introduction section (lines 53 to 55). An earlier study proposing this concept should be cited (Arch Neurol 2002; 59: 1771-6).
6. For a sentence “The misfolded monomers aggregates generating amyloid fibrils, which precipitate into tissues” (lines 52 to 53), more recent article demonstrating this phenomenon should be cited here (Molecules 2021; 26: 5091).
7. Please reconfirm the format of references.
Author Response
Dear Editors and Reviewers,
thanks for your comments. We would like you will evaluate our revised version of the manuscript for possible publication in Biomedicines.
Reviewer #2:
The authors conducted bioelectrical impedance analysis in 15 patients with hereditary transthyretin (ATTRv) amyloidosis before and 9 months after the administration of patisiran, one of the novel RNA interfering agents. The body composition significantly changed following 9 months of treatment.
This is an important study providing important insights into current knowledge on the treatment of ATTRv amyloidosis. Taking up the topic of ATTRv amyloidosis is timely because novel disease-modifying therapies, such as transthyretin stabilizers, RNA interfering agents, antisense oligonucleotides, and gene editing agents, now appear one after another. Therefore, this manuscript will attract broad range of readers. It is well written, and I do not have any critical comments.
We thank the reviewer for these positive comments giving us the opportunity to improve our manuscript. We really appreciate these encouraging considerations; we are doing our best to improve our manuscript according to the reviewer’s comments.
Minor issues to strengthen this manuscript are raised as follows:
- Please reconfirm the use of abbreviations. For example, “HGS” in the abstract should be spelled out.
We thank the reviewer for this punctual correction. We added the explanation of abbreviation the first time used.
- Although the authors stressed hand grip strength for the monitoring the disease progression and response to treatment as well as bioelectrical impedance analysis, the values did not significantly change after the treatment. It seems to be better to tone down the issue of hand grip strength because the results of bioelectrical impedance analysis are sound.
We thank the reviewer for this punctual observation. We modified abstract and conclusions reducing emphasis on HGS at difference with BIA.
- Is this a retrospective study, or a prospective one?
It is a prospective study. We specified it in the methods.
- I would suggest briefly stating the ethics issue in the methods section although it is incorporated in the last part of the main text.
We added ethics in the methods section.
- The authors mentioned the issue of early-onset Val30Met cases from endemic foci and late-onset cases from non-endemic areas in the first paragraph of the introduction section (lines 53 to 55). An earlier study proposing this concept should be cited (Arch Neurol 2002; 59: 1771-6).
We thank the reviewer for this punctual suggestion. We added this precious study to the references on Val30Met cases.
- For a sentence “The misfolded monomers aggregates generating amyloid fibrils, which precipitate into tissues” (lines 52 to 53), more recent article demonstrating this phenomenon should be cited here (Molecules 2021; 26: 5091).
We thank the reviewer for this punctual suggestion. We also added this precious review article.
- Please reconfirm the format of references.
Thank you for your suggestion. We have revised the reference list.
Hoping in positive feedback we look forward to hearing from you soon.
Kind regards,
Vincenzo Di Stefano